# Effect of Ramadan Fasting on Weight and Body Composition in Healthy Non-Athlete Adults: A Systematic Review and Meta-Analysis

**DOI:** 10.3390/nu11020478

**Published:** 2019-02-24

**Authors:** Hamish A. Fernando, Jessica Zibellini, Rebecca A. Harris, Radhika V. Seimon, Amanda Sainsbury

**Affiliations:** Faculty of Medicine and Health, Nutrition, Exercise and Eating Disorders, The Boden Institute of Obesity, The University of Sydney, Charles Perkins Centre, NSW 2006, Australia; hf.hamishfernando@gmail.com (H.A.F.); jzib2585@uni.sydney.edu.au (J.Z.); rebecca.harris@sydney.edu.au (R.A.H.); radhika.seimon@sydney.edu.au (R.V.S.)

**Keywords:** Ramadan, intermittent fasting, body weight, body composition

## Abstract

Background: Ramadan involves one month of fasting from sunrise to sunset. In this meta-analysis, we aimed to determine the effect of Ramadan fasting on weight and body composition. Methods: In May 2018, we searched six databases for publications that measured weight and body composition before and after Ramadan, and that did not attempt to influence physical activity or diet. Results: Data were collected from 70 publications (90 comparison groups, 2947 participants). There was a significant positive correlation between starting body mass index and weight lost during the fasting period. Consistently, there was a significant reduction in fat percentage between pre-Ramadan and post-Ramadan in people with overweight or obesity (−1.46 (95% confidence interval: −2.57 to −0.35) %, *p* = 0.010), but not in those of normal weight (−0.41 (−1.45 to 0.63) %, *p* = 0.436). Loss of fat-free mass was also significant between pre-Ramadan and post-Ramadan, but was about 30% less than loss of absolute fat mass. At 2–5 weeks after the end of Ramadan, there was a return towards, or to, pre-Ramadan measurements in weight and body composition. Conclusions: Even with no advice on lifestyle changes, there are consistent—albeit transient—reductions in weight and fat mass with the Ramadan fast, especially in people with overweight or obesity.

## 1. Introduction

Ramadan falls on the ninth lunar month of the Islamic Hijra calendar, and is considered to be the holiest month of the Islamic religion. With the world population of Muslims just under 2 billion in 2010 [1], there are hundreds of millions of people that practice the commitments of Ramadan, which include abstinence from eating and drinking from sunrise to sunset. This could mean that, depending on location, Ramadan fasting time could range from 9 to 22 hours per day [2,3].

Effects of Ramadan fasting on weight vary between individuals, ranging from weight loss to weight gain, depending on whether or not energy intake in the non-fasting period under- or over-compensates for the lack of energy intake during the fasting period [4]. A meta-analysis that used data from 35 publications showed that Ramadan fasting caused a statistically significant decrease in weight (−1.24 kg by the end of Ramadan, −1.51 kg in men and −0.92 kg in women), while the weight loss observed at follow-up (2–6 weeks after Ramadan) was less pronounced, albeit weight was still statistically significantly lower compared to before Ramadan (−0.27 kg) [5].

While the overall effects of Ramadan fasting on weight are now well-known, its overall effects on body composition (i.e., fat mass and fat-free mass) are not clear. Ramadan fasting could conceivably affect body composition, because it has been shown to affect physical performance or activity. Indeed, in sedentary individuals, heart rate and ventilator responses to moderately-intense aerobic exercise were significantly less during Ramadan fasting compared to before, indicating a drop in performance [6]. Reduced performance has also been shown to occur in well-trained individuals: as examples, Ramadan fasting has been shown to have negative effects on muscular performance in fighter pilots [7], and to reduce physical performance scores in young men in the army [8]. Such changes in physical function during Ramadan fasting could potentially lead to larger than desirable losses of fat-free mass relative to the loss of fat mass [8]. Understanding how Ramadan fasting affects both weight as well as body composition of populations, on average, would lead to a better understanding of how Ramadan affects the global Muslim community. This could be beneficial considering the epidemic of overweight and obesity, which affects countries around the world, including Muslim-majority countries [9,10]. Although there have been studies that have analysed body composition changes with Ramadan fasting, the results have been variable with no clear consensus.

In light of this, the aim of the present systematic review and meta-analyses was to assess how Ramadan fasting affects weight and body composition in adults. We selected publications that investigated non-athletes, and that did not attempt to influence physical activity or diet in participants, in order to reflect Ramadan fasting as implemented by most people.

## 2. Materials and Methods

### 2.1. Search Strategy

The literature search was conducted across six databases: Medline, Premedline, Embase, Scopus, Cinahl, and Global Health. We searched for publications addressing the effects of Ramadan fasting on weight and/or body composition. Free-text terms used for the literature search were as follows (based on keywords): “Ramadan” AND (“weight” OR “body composition” OR “fat mass” OR “fat percentage”). The search term “weight” included 98.6% of the literature with body composition analyses (data shown in results). The search acquired publications in any language, from database inception up to May 2018. Reference lists of relevant publications and reviews were searched to help ensure that all relevant publications were found.

### 2.2. Inclusion and Exclusion Criteria

Only original research published in peer-reviewed scientific journals, which measured changes in the parameters of interest in healthy non-athlete adults at a minimum of two of the three time points of interest (see data extraction and organization), were included in this work. Parameters of interest were weight, body composition i.e., fat mass as a percentage of weight (‘fat percentage’, %), absolute fat mass (kg), or fat-free mass (kg). As Ramadan fasting has been shown to affect aspects of physical activity [6,7,8], and as physical activity is known to affect body composition (our primary outcome of interest), we also extracted available data on physical activity from the included publications. We excluded studies involving participants with acute/chronic disease, on medications, or that were under 16 years of age, pregnant, breastfeeding, or athletes. Studies were also excluded when the protocol explicitly aimed to influence the physical activity or diet of participants in any way during Ramadan fasting, or when the publication did not give specific values for weight or body composition of participants, when identical data was published more than once (in which case the newer publication(s) was/were excluded), lack of availability of an English translation of the full text of the publication, and no availability of a full text. If a broad statement was used to describe any of the time points, for example, “just before Ramadan” or “end of Ramadan”, instead of a clearly defined time point, the study was still included.

### 2.3. Screening of Publications

Screening of publications identified in the search was independently conducted by two authors, HAF and JZ, based on the inclusion and exclusion criteria mentioned above. Screening was conducted by first going through all titles and abstracts in order to exclude publications that were clearly irrelevant. Thereafter, full texts of the remaining publications were retrieved and screened in order to further assess whether the studies therein fit our selection criteria. When there were discrepancies between the two authors about which studies to include, consensus was reached by discussion between the two authors, without the need for consultation with an additional author.

### 2.4. Data Extraction and Organization

The following data were extracted from each included study and tabulated (Appendix A): publication first author and year of publication, sample size (male (M), female (F), total), mean and standard deviation (or range) of the age and body mass index (BMI) of participants before Ramadan, year study was conducted (which may have been different from the year the study was published), the location where the study was conducted, fasting duration rounded to the nearest half an hour (note that if fasting duration was not stated in the publication, but the year and location in which the study was conducted were given, fasting duration was estimated based on the average time between sunrise and sunset during Ramadan in the relevant year and location), timing of the pre-Ramadan measurement (‘pre-R’), timing of the post-Ramadan measurement (‘post-R’), timing of the follow-up measurement (‘follow-up’) relative to the Ramadan fast, mean and standard deviation of weight, method of body composition measurement, mean and standard deviation of fat percentage (%), absolute fat mass (kg), and fat-free mass (kg), method of physical activity measurement, mean and standard deviation of ‘maximum effort physical activity’ (which involved some type of exercise-related measurement of a maximum effort over a certain time, e.g., the maximum weight that could be pushed in a leg press in a single repetition, or the maximum oxygen uptake when running on a treadmill), and ‘daily physical activity’ (measurement of the total amount of a certain type of physical activity undertaken across the day, e.g., total steps taken). Weight measurements reported in pounds were converted to kilograms, and all times reported in weeks or months were converted to days (7 days a week, ~30 days a month).

If any study had multiple comparison groups (i.e., female versus male, overweight/obese versus normal weight), the separate comparison groups were included as separate data points in the meta-analyses.

Corresponding authors were contacted if any required data were unavailable in a publication, or were published in a format different from that required for our meta-analyses. Only if there was a response with the appropriate data was the study included in the meta-analyses.

### 2.5. Analyses and Subgroups

Analyses were done to determine the change in the selected parameter between pre-R and post-R, and between pre-R and follow-up.

After collecting the data on demographics in Appendix A, we noticed distinct sub-populations that were of particular interest for analysis. Thus, analyses of weight change were conducted using subgroups based on BMI category at pre-R (overweight/obese and normal weight) and sex (female and male). Analyses of fat percentage, absolute fat mass and fat-free mass were also conducted using subgroups based on sex, and analyses of fat percentage were also divided into subgroups based on BMI (there was inadequate data available for other body composition parameters for subgrouping based on BMI). Overweight and obesity were grouped together in a single subgroup instead of separately due to there being few studies in which each category of BMI was investigated separately [11,12,13,14,15,16,17,18], and also due to there being variations between studies in the criteria used for BMI categories. Four studies used the World Health Organization criteria for BMI categorization (18.5 kg/m^2^ ≤ BMI < 25 kg/m^2^ = normal weight; 25 kg/m^2^ ≤ BMI < 30 kg/m^2^ = overweight; BMI ≥ 30 kg/m^2^ = obese) [16,18,19,20], two studies used the National Institutes of Health criteria (BMI ≤ 27.3 kg/m^2^ = normal weight; BMI > 27.3 kg/m^2^ = obese) [12,17], and three studies did not specify the criteria used [11,13,15]. Due to the variety of criteria used for the classification of BMI categories, we categorized BMI according to the categorization used in each study, irrespective of the criteria used.

The final subgroup analysis conducted was weight change by geographical location, to assess whether this had any effect on the weight and body composition outcomes of the Ramadan fast. Areas of similar cultural backgrounds were grouped together, hence the four subgroups used in this review were: Middle East + North Africa; South Asia; South East Asia; and Westernized countries.

### 2.6. Data processing, Meta-Analyses, and Meta-Regressions

In order to determine the effect of Ramadan fasting on the various parameters, measurements obtained at pre-R were subtracted from those obtained at post-R or follow-up for each study included in the analyses.

In order to perform the meta-analyses, for which a random-effects model was used, changes in means between different time points (pre-R versus post-R, and pre-R versus follow-up), as well as the standard deviations of the changes, were required. However, most of the studies included in this work did not provide standard deviations of the changes; only standard deviations of the mean values at each time point. Therefore, we employed an imputation method to determine standard deviations of the changes between time points, as outlined in the Cochrane Handbook [21]. In brief, standard deviation of change was imputed with the equation
SD_change_ = √[SD_baseline_^2^ + SD_final_^2^ − (2 × Corr × SD_baseline_ × SD_final_)](1)
where SD_change_ = standard deviation of the change between time points; SD_baseline_ = standard deviation of ‘baseline’ mean (in this study, pre-R or post-R); SD_final_ = standard deviation of ‘final’ mean (in this study, post-R or follow-up); Corr = Correlation coefficient between the ‘baseline’ standard deviation and the ‘final’ standard deviation.

For the purpose of this study, publications that included the standard deviation of the change of the relevant variable were used to calculate the correlation coefficient between the ‘baseline’ standard deviation and the ‘final’ standard deviation for each of those studies, using the above formula. The average correlation coefficient from these publications was then used to calculate the standard deviation of change for all studies in the analyses. For weight, there were five relevant publications that reported the standard deviation of the change between pre-R and post-R [22,23,24,25,26], while standard deviations of change of other variables of interest, that is, fat percentage, absolute fat mass and fat-free mass, were found within two publications [24,26] between pre-R and post-R. The standard deviation between pre-R and follow-up were found in only one publication [24]. As the correlation coefficients between the ‘baseline’ standard deviation and the ‘final’ standard deviation were similar for weight and body composition parameters between both pre-R and post-R (~0.85), as well as pre-R and follow-up (~0.75), the average correlation coefficient calculated for weight between pre-R and post-R (~0.80) was used for imputation of the standard deviation of change for both weight and body composition parameters between both sets of time points.

In addition to meta-analyses and subgroup analyses, meta-regression analyses were performed to examine any potential correlation between BMI at pre-R and weight change between pre-R and post-R, as well as any potential correlation between fasting duration and weight change between pre-R and post-R. For the correlation between fasting duration and weight change, two meta-regression analyses were performed: one based only on studies which reported the average daily fasting duration in the publication, and the other also including studies that did not explicitly mention the average daily fasting duration, but from which fasting duration was inferable based on the year and location of the study (since Ramadan fasting occurs from sunrise to sunset) [27].

No publication bias was detected in any of our meta-analyses, when investigated with funnel plots and Egger’s tests. Due to the nature of our meta-analyses, where data were compared before and after Ramadan fasting, with no randomization nor any independent grouping, the usual quality filters that apply to randomized controlled trials or observational studies could not be applied.

All statistical analyses were conducted using Comprehensive Meta-Analysis (CMA) software version 3.3.070 (Biostat, Englewood, NJ, USA). Statistical significance was set at *p* < 0.05 for all analyses.

## 3. Results

As shown in Figure 1, the search strategy identified 1153 publications, 398 of which were duplicates, resulting in 755 unique publications. The titles and abstracts of these publications were then screened against the inclusion and exclusion criteria, resulting in the exclusion of 688 publications and the retention of 67. Full texts of these 67 publications were retrieved and screened. As all of these 67 publications met our criteria, they were included. Three more publications were found through the reference lists of these 67 publications, resulting in 70 publications being included in this work [11,12,13,14,15,16,17,18,19,20,22,23,24,26,28,29,30,31,32,33,34,35,36,37,38,39,40,41,42,43,44,45,46,47,48,49,50,51,52,53,54,55,56,57,58,59,60,61,62,63,64,65,66,67,68,69,70,71,72,73,74,75,76,77,78,79,80,81,82]. These 70 publications are listed (in order of publication date) in Appendix A. Of the 70 publications included, 69 (98.6%) were targeted by our search term “weight”, indicating that this search term was an umbrella term for publications that also measured body composition, demonstrating that these two outcomes are usually measured together. These 70 publications included 90 comparison groups and 2947 participants. Of these 90 comparison groups, the present analyses used 89 that measured weight, 24 that measured fat percentage, 20 that measured absolute fat mass, 19 that measured fat-free mass, 3 that measured maximum effort physical activity, and 3 that measured daily physical activity.

Sample sizes of the included comparison groups, shown in Appendix A, varied from 7 to 240 participants, ages ranged from 16 to 70 years, and the reported mean BMI of participants at the pre-R measurement ranged from 21.3 to 39.7 kg/m^2^. The studies were conducted between 1982 and 2018, and fasting durations varied from 10 to 17 hours. Timing of the measurements made pre-R varied from 21 days before Ramadan to day 1 of Ramadan, and post-R measurement timing varied from day 14 of Ramadan to 7 days after the end of Ramadan, while follow-up measurement timing was between 14 to 35 days after the end of Ramadan.

To briefly explain the format of figures containing forest plots (e.g., Figure 2), the summary results for each meta-analysis are shown in numbers at the bottom of the four columns of numbers. For subgroup analyses (by sex, starting BMI category, and location), the summary results for each subgroup are additionally shown at the bottom of each subgroup. The position of each square in the forest plot illustrates the effect size (difference in means between time points) for each comparison group, and the lines extending horizontally from each square illustrate the 95% confidence interval (CI) for each study. The percentage weight contributed by each study to the overall meta-analysis is shown by the area of the effect size square (the greater the percentage weight contributed by a study, the smaller the square). The diamond at the bottom of each forest plot, or subgroup within a forest plot, depicts the summary result for each meta-analysis (effect size), and the width of the diamond shows the 95% CI for the summary effect size.

Heterogeneity statistics are shown in the legend of each figure containing a forest plot. Tau (*T*) gives the standard deviation of the true effect size, while *p* denotes the significance of *T*. A significant *p* value for *T* indicates that the true effect size is variable. It is important to note, however, that an insignificant *p* value does not definitely imply that the true effect size has low variability, especially when small studies or studies with large variances are used in the meta-analysis, as these factors lower the statistical power of the analysis. Furthermore, the *p* value for *T* does not give any indication of the amount of variability in the true effect size, only the significance. Finally, *I*^2^ indicates what proportion of the observed variation in effect sizes between studies is due to variability in the true effect size [83]. In other words, *I*^2^ tells us what proportion of the observed variability between studies would remain if variability due to sampling error were removed. For instance, an *I*^2^ of 0% means that all of the observed differences between study effect sizes are due to sampling error; an *I*^2^ of 100% means that all of the observed differences between studies are due to variability in the true effect. As a rule of thumb, *I*^2^ values of 25, 50, and 75% are considered low, moderate and high, respectively [84]. It is important to note that *I*^2^ does not denote anything about the amount of variability in the true effect size. While heterogeneity statistics were calculated for all analyses, they are quoted in the text only where relevant, for example, when the result was a highlight of the paper and/or when a result may have been influenced by significant heterogeneity. When both a significant *p*-value for *T* (*p* < 0.05) and a high value of *I*^2^ (*I*^2^ > 75%) was observed for an analysis, the analysis was repeated after removing outliers, unless specifically stated otherwise.

### 3.1. Weight Changes

#### 3.1.1. Ramadan Fasting Promoted a Transient Reduction in Weight

When assessing the change in weight from pre-R to post-R, it is apparent that weight decreased significantly (−1.34 (95% CI: −1.61 to −1.07) kg, *p* < 0.001) (Appendix A). By follow-up, there was still a significant decrease in weight compared to pre-R (−0.59 (−0.99 to −0.20) kg, *p* = 0.003) (Appendix A).

#### 3.1.2. Starting BMI may have Influenced Weight Changes in Response to Ramadan Fasting

Appendix A indicates that people in both the overweight/obese subgroup as well as in the normal weight subgroup lost a significant amount of weight between pre-R and post-R (−3.53 (−5.47 to −1.60) kg in overweight/obese, *p* < 0.001, versus −2.06 (−3.37 to −0.74) kg in normal weight, *p* = 0.002). Despite the significant decrease in weight from pre-R to post-R in both overweight/obese and normal weight subgroups, weight change was not statistically significantly different from pre-R to follow-up in either group (Appendix A). There was no significant difference between overweight/obese and normal weight subgroups between either of these two sets of time-points (Appendix A).

The subgroup analyses were limited by the small number of studies with participants in specific BMI categories (11 which compared pre-R and post-R; 5 which compared pre-R and follow-up). In contrast, there were 33 studies which reported baseline BMI, and so we conducted meta-regression between pre-R BMI (a continuous variable) and weight change between pre-R and post-R to further explore the link between BMI and Ramadan fasting effects. As shown in Appendix A, there was a statistically significant correlation between these two parameters (*p* = 0.045), demonstrating that the greater the BMI prior to Ramadan fasting, the greater the weight loss.

#### 3.1.3. Sex did not Influence Weight Changes in Response to Ramadan Fasting

In the case of both female and male subgroups, weight decreased significantly by the end of the Ramadan fast (Appendix A), however 2–5 weeks after Ramadan there was no longer any significance in the change in either group (Appendix A), and there was no significant difference between female and male subgroups.

#### 3.1.4. Location may have Influenced Weight Change in response to Ramadan Fasting

To determine whether geographical location affected the outcome of Ramadan fasting, subgroup analyses were performed using four locations: Middle East + North Africa; South Asia; South East Asia; and Westernised countries. As shown in Appendix A, the decrease in weight from pre-R to post-R was significant for the Middle East + North Africa (−1.22 (−1.43 to −1.01) kg, *p* < 0.001), South Asia (−1.11 (−2.04 to −0.18) kg, *p* = 0.019) and South East Asia (−3.68 (−5.67 to −1.68) kg, *p* < 0.001), but not in Westernized countries (−0.91 (−1.97 to 0.15) kg, *p* = 0.091). The removal of outliers due to the high degree of heterogeneity (significant *T* value and *I*^2^ > 75%) observed in the South East Asia subgroup had no significant effect on the outcome. By follow-up, the decrease in weight relative to pre-R was significant only for the Middle East + North Africa subgroup (−0.62 (−1.08 to −0.16) kg, *p* = 0.009) (Appendix A). However, it is important to note that these differences could be brought about by the differences in the number of studies in each subgroup. Overall, it appears that the location of fasting has an influence on weight change with Ramadan. If this is true, then variation in fasting duration is one factor that could potentially influence this.

A regression analysis was conducted between the average number of hours scheduled for fasting per day and weight change between pre-R and post-R. Appendix A show that there was no significance in the correlation between scheduled fasting duration and Ramadan-induced weight change. This indicates that variations in the designated duration of daily fasting during Ramadan, which is dependent on location and the time of year in which the fast takes place, does not have an influence on weight change.

### 3.2. Body Composition Changes

#### 3.2.1. Ramadan Fasting Promoted a Transient Reduction in Fat Mass as a Percentage of Weight

When assessing overall changes in body composition across Ramadan, fat percentage decreased significantly from pre-R to post-R (−1.07 (−1.55 to −0.59) %, *p* < 0.001) (Figure 2). By follow-up, however, there was no significant change in fat percentage from pre-R (Appendix A), indicating a return to pre-R fat percentage after the initial fat loss. There was significant heterogeneity in the analyses for fat percentage change between pre-R and post-R, as well as pre-R and follow-up, with a large proportion of this variability being due to variability in true effect size (pre-R versus post-R: *T* = 0.829, *p* < 0.001, *I*^2^ = 61.5%; pre-R versus follow-up: *T* = 2.083, *p* < 0.001, *I*^2^ = 89.7%). The removal of outliers due to the high degree of heterogeneity (significant *T* value and *I*^2^ > 75%) observed between pre-R and follow up had no significant effect on the outcome.

Subgroup analyses revealed that the overweight/obese subgroup exhibited a statistically significant decrease in fat percentage between pre-R and post-R (−1.46 (−2.57 to −0.35) %, *p* = 0.010), while the normal weight subgroup did not (−0.41 (−1.45 to 0.63) %, *p* = 0.436) (Figure 3). The difference in change in fat percentage between subgroups was not significant. Due to the high degree of heterogenity observed in the overweight/obese subgroup (*T* = 1.240, *p* < 0.001, *I*^2^ = 86.8%), the analysis was repeated after removal of outliers, and the decrease in fat percentage from pre-R to post-R was still statistically significant, and there was still no statistically significant difference from the results in people of normal weight. By follow-up at 2–5 weeks after Ramadan, however, neither subgroup showed a significant change from pre-R in fat percentage, albeit in the overall group there was still a small but statistically significant reduction in fat percentage (−0.94 (−1.73 to −0.15) %, *p* = 0.020) (Appendix A).

The final subgroup analysis for fat percentage was conducted by sex. In this case, both female and male subgroups showed a significant decrease in fat percentage from pre-R to post-R: (−0.62 (−1.00 to −0.24) %, *p* = 0.001 for the female subgroup and −0.86 (−1.22 to −0.50) %, *p* < 0.001 for the male subgroup) (Figure 4). The difference in change in fat percentage between female and male subgroups was not significant. The change in fat percentage from pre-R values in both subgroups was not statistically significant when measured at follow-up (Appendix A).

#### 3.2.2. Ramadan Fasting Promoted a Transient Reduction in Absolute Fat Mass

Ramadan fasting led to a significant decrease in absolute fat mass when measured at post-R (−0.98 (−1.05 to −0.92) kg, *p* < 0.001) (Figure 5), although no significant decrease was seen when measured at follow-up (Appendix A), indicating a return to pre-R absolute fat mass after the initial fat loss.

Due to the relatively small number of comparison groups that reported absolute fat mass (Figure 5 and Appendix A), no subgroup analyses could be conducted on the basis of BMI category at pre-R. However, subgroup analyses were performed on the basis of sex (Figure 6), and these revealed that absolute fat mass decreased significantly in both females (−0.67 (−1.29 to −0.06) kg, *p* = 0.032) and males (−0.99 (−1.05 to −0.93) kg, *p* < 0.001) by post-R. There was no significant difference in absolute fat mass between pre-R and follow-up in either female or male subgroups (Appendix A). In both of these timescales (pre-R to post-R, and pre-R to follow-up), there was no significant difference in the change in absolute fat mass between the female and male subgroups.

#### 3.2.3. Ramadan Fasting Promoted a Transient Reduction in Fat-Free Mass

Similar patterns of change to those seen for absolute fat mass were observed for absolute fat-free mass. Ramadan fasting caused a significant decrease in absolute fat-free mass when measured at post-R (−0.66 (−0.75 to −0.57) kg, *p* < 0.001) (Figure 7), although the difference in fat-free mass between pre-R and follow-up was not statistically significant (Appendix A), indicating a return to pre-R fat-free mass after the initial loss of fat-free mass. Loss of fat-free mass (−0.66 kg, Figure 7) was approximately 30% less than the loss of absolute fat mass (−0.98 kg, Figure 5).

Due to the relatively small number of comparison groups that reported fat-free mass (Figure 7), no subgroup analyses of fat-free mass could be conducted on the basis of BMI category at pre-R. However, subgroup analyses based on sex (Figure 8) showed that although males lost a significant amount of fat-free mass from pre-R to post-R (−0.69 (−0.79 to −0.60) kg, *p* < 0.001), females did not (−0.21 (−0.56 to 0.15) kg, *p* = 0.253). Between pre-R and follow-up, there was no significant change in fat-free mass from pre-R in female or male subgroups (Appendix A). In both timescales, there was no significant difference between female and male subgroups with respect to the change in fat-free mass.

### 3.3. Physical Activity

This study included two types of physical activity measurements: maximum effort physical activity, which incorporated measures of maximal efforts typically during exercise, and daily physical activity, which incorporated measurements estimating the overall amount of a physical activity in a day (such as total number of steps). Due to the variation in physical activity measures used, a standardised mean difference (Hedges’ g) was used for this analysis, instead of a mean difference. The meta-analyses of measures of physical activity did not indicate any significant change between pre-R and post-R of either maximum effort physical activity (Appendix A) or daily physical activity (Appendix A). There was insufficent data to conduct analyses of the changes in physical activity between pre-R and follow-up.

## 4. Discussion

This systematic review and meta-analysis shows that on the whole, Ramadan fasting led to statistically significant reductions in weight and all parameters of body composition (i.e., fat mass, both as a percentage of weight (%) or as absolute mass (kg), and fat-free mass (kg)). Starting BMI was significantly and positively correlated with weight loss during the Ramadan fast (i.e., the greater the starting BMI, the more weight that was lost). This suggests that Ramadan fasting may be more effective—in terms of weight loss—in people with a higher BMI. In keeping with this, the reductions in weight and fat percentage were of greater magnitude in people with overweight/obesity versus those of normal weight, with the reduction in fat percentage being significant only in people with overweight/obesity and not in those of normal weight. The loss of fat-free mass during the Ramadan fast appeared to be about 30% less than the loss of absolute fat mass, consistent with the observed reduction in fat mass as a percent of weight. All of these changes in weight and body composition occurred despite the fact that none of the studies in this systematic review and meta-analysis included an intervention to alter physical activity or diet. Moreover, Ramadan fasting did not appear to change physical activity. By 2–5 weeks after Ramadan fasting, weight was still significantly reduced compared to pre-Ramadan, but the weight loss was of a lesser magnitude than just after Ramadan. All body composition parameters had returned to baseline values by this time. Thus, while Ramadan may represent an opportunity for weight and fat loss for people with overweight or obesity, work is required to identify strategies for long-term maintenance of reduced weight and fat mass.

Weight loss with a greater proportion being from absolute fat mass and a lesser proportion being from fat-free mass is considered ideal, due to the adverse effects of fat mass on health [85], and the role of fat-free mass in maintaining skeletal integrity and functional capacity, resting energy expenditure and glucose homeostasis [86]. According to the present meta-analyses, over half of the weight lost during Ramadan fasting may be fat mass. A systematic review indicated that loss of fat-free mass could account for 4.3 to 38.3% of weight lost during obesity interventions based on diet and exercise [87]. Taken together, these insights suggests that the amount of fat-free mass lost during Ramadan fasting as shown in the current study is approximately within an expected range.

Most of the studies included in the current meta-analyses of body composition estimated it either with skin calipers or with bioelectrical impedance. Both of these methods use a two-compartment model for estimating body composition, and both have been shown to be relatively inaccurate [88,89,90,91,92,93]. Since only three studies in our meta-analyses used more accurate equipment that measures body composition using the three-compartment model of hydrodensitometry [31,51,73], it was not possible to conduct subgroup analyses using only the studies that utilised this more accurate measurement method without compromising statistical power. However, while inaccurate in showing changes in body composition in an individual person, bioelectrical impedance methods may be reasonably accurate in measuring average body composition of groups of individuals [94,95]. Furthermore, one study suggested that the inaccuracy of bioelectrical impedance methods to measure changes in body composition over time was exacerbated by faster weight loss, although there were a few caveats to this study, including a small sample size and some variable findings [96]. However, if future studies verify this finding, then bioelectrical impedance methods may provide relatively accurate assessment of changes in body composition during the Ramadan fast, since the weight changes during Ramadan were relatively small and slow (being an average of 1.36 kg over the month).

The overall decrease in weight observed between pre-R and post-R (1.36 kg), as well as the significant weight loss shown in both females and males, is similar to the findings of a previous systematic review and meta-analysis [5]. The current study extends those previous findings by demonstrating that not only is the weight lost with Ramadan fasting associated with significant loss of fat (and fat-free) mass, but also that the loss of weight and fat mass could possibly be influenced by starting BMI. Indeed, people with overweight/obesity lost more weight during Ramadan than people of healthy weight (3.53 kg versus 2.06 kg, respectively), albeit this difference was not statistically significant, perhaps related to high heterogeneity among the relatively smaller number of studies that investigated people in specific BMI categories. There was, however, a statistically significant and positive correlation between BMI and weight loss (the greater the BMI before Ramadan fasting, the greater the amount of weight lost), and fat percentage was significantly reduced by the end of Ramadan in people with overweight/obesity but not in people of normal weight. One possible reason for this greater weight loss is that people with greater BMIs are likely to carry more body water due to greater glycogen stores than people of normal weight, and hence would be expected to lose more fluid in response to fasting [97]. Furthermore, it has been established that during weight loss, people of normal weight lose a greater proportion of weight as fat-free mass than people with overweight/obesity—in excess of 35% versus 20 to 30%, respectively [98]—thus potentially resulting in more favourable fat percentage changes in individuals with higher BMIs.

Our work suggests that location may influence weight changes during Ramadan fasting in a similar way to that reported in a previous meta-analysis [5]. The differences in weight lost during Ramadan between the different regions could possibly be explained via cultural influences on food habits (e.g. the energy content of the food typically consumed in each region). Another way in which location could conceivably influence weight changes during Ramadan is via the duration of fasting. However, our meta-regression showed that fasting duration did not influence weight change during Ramadan.

Although our systematic review and meta-analyses excluded studies involving interventions that sought to alter physical activity or diet, we were interested to know if participants exhibited changes in physical activity (or energy intake) during Ramadan fasting. Our meta-analyses showed that there was no apparent change in physical activity, albeit there was very little data available (six studies: three of which measured maximum effort physical activity, and three of which measured daily physical activity). There was also no clear-cut reduction in energy intake during Ramadan fasting. Indeed, of the 13 publications (and 14 comparison groups) that reported statistics for the analysis of energy intake before and during Ramadan fasting, 7 reported no significant difference in energy intake between time points [19,24,33,46,50,54,61], 4 reported a significant increase in energy intake during Ramadan [20,31,34,39], and only 2 reported a significant decrease in energy intake during Ramadan [38,43]. However, all of the studies in these publications required participants to self-report their energy intake, which is known to be highly erroneous [99,100].

These findings of no apparent change in physical activity or energy intake (albeit self-reported), and in the absence of any intervention, lead to the question of how the Ramadan fast induced weight or fat loss at all. One possibility could be an increase in energy expenditure. Indeed, time-restricted feeding (restricting eating time to only eight hours per day) in mice resulted in an increase in energy expenditure and protection against diet-induced obesity relative to ad libitum-fed mice, despite consumption of a high fat diet and no reduction in energy intake [101]. While Ramadan fasting might be considered a form of time-restricted feeding, albeit with eating times somewhat different from usual eating times, previous studies have indicated only minor, non-significant changes in energy expenditure in response to Ramadan fasting [33,58,102,103]. The methods used for the measurement of energy expenditure in at least two of these studies—armband monitor [102] and tri-axial accelerometer (worn on hip) [58]—although reliable in general, are not free of inconsistencies and inaccuracies [104,105]. Another of these studies was only a small pilot study involving 12 individuals, 5 of whom had diabetes [103]. Hence, although the data indicate that weight loss during Ramadan fasting occurred despite little or no change in physical activity, energy intake or energy expenditure, further research using more accurate approaches to measurement may be required to confirm this.

As mentioned above, given that the overall weight and fat mass loss that was observed in the studies included in our meta-analyses, especially in people with overweight or obesity, occurred with no explicit instructions or advice to alter physical activity or diet, Ramadan fasting appears to be a potential opportunity for intervention in obesity. However, the post-Ramadan weight regain could be an obstacle when considering longer-term, sustained weight management. According to our findings, there was a return to, or towards, baseline values within five weeks after the end of Ramadan for all measured parameters. This indicates that any beneficial changes in weight and/or body composition observed by the end of Ramadan were transient. However, periodic fasting in some form after the Ramadan period appears to be a promising option for preventing weight regain post-Ramadan, especially when the fasting is based on religious principles and faith compared to standard dietary guidelines alone [106]. Furthermore, even if post-Ramadan weight regain occurs, recent studies have suggested that even transient decreases in weight may lead to positive long-term health outcomes [107,108].

In addition to loss of weight as shown in a previous [5] and the present meta-analysis, as well as the loss of fat shown in the present meta-analysis, Ramadan fasting also has metabolic benefits, as apparent in three recent systematic reviews [109,110,111], one of which included a meta-analysis [111]. These studies revealed that the benefits of the Ramadan fast include greater circulating levels of high-density lipoprotein cholesterol (HDL-C), with concomitant reductions in blood pressure and circulating levels of low-density lipoprotein cholesterol (LDL-C), total cholesterol and glucose. These changes were shown in healthy participants as well as in patients with chronic illnesses or disorders such as the metabolic syndrome. The metabolic benefits appeared particularly pronounced in males [111]. There have, however, been mixed results when assessing the impact of Ramadan fasting on people with type 2 diabetes mellitus, especially in terms of the profile of lipids in the circulation. Although beneficial in the case of many parameters including reduced fat mass [103] and improved glycaemic control [112], several studies have shown unfavourable effects of Ramadan fasting on lipid profiles in people with diabetes [112,113,114,115], albeit there has also been one large-scale study [116] and one small study [103] that showed a neutral effect on lipid profile. As such, more research into the effects of Ramadan fasting on people with diabetes is needed, and care needs to be taken when people with diabetes follow the Ramadan fast.

There are a few strengths and limitations that need to be highlighted with regard to the current systematic review and meta-analysis. The review was conducted according to PRISMA guidelines (Preferred Reporting Items for Systematic Reviews and Meta-Analyses) [117], as shown in Appendix A. With regard to limitations, the participants in the studies used for the current meta-analyses were not randomly selected, but were instead people who intended to undertake Ramadan fasting, and this leads to selection bias, which may also reduce generalizability of our findings. Moreover, when weight changes were analysed by starting BMI category, subgrouping was done according to the BMI category stated in each included study, despite there being variability in the criteria used for the categorisation of BMI in different studies. This meant that there may have been some overlap in the BMI ranges that constituted overweight/obese and normal weight between different studies. Finally, as mentioned above, results for body composition changes may not be reliable, as the methods most frequently used to track these changes were based on two-compartment models, which are relatively inaccurate. It is possible however that the slow weight loss characteristic of the Ramadan fast may have reduced the impact of this inaccuracy, although this would need to be further explored.

## 5. Conclusions

In conclusion, Ramadan fasting led to significant decreases in weight, fat mass (as a percentage of weight and as absolute mass), and fat-free mass. The greater the starting BMI, the greater the amount of weight lost. Moreover, people with overweight or obesity lost a significant amount of fat mass (as a percentage of weight), and this effect of Ramadan fasting was not seen in people of normal weight. Since these reductions in weight and fat mass occurred despite excluding studies where any kind of physical activity directives, diet plan, or other lifestyle coaching were given, these findings imply that with encouragement to exercise, as well as dietary guidance and support, not only during the Ramadan fasting period but also post-Ramadan, the positive effects achieved during Ramadan could be enhanced, and the return to pre-Ramadan weight and parameters of body composition could possibly be prevented. Taken together, these findings suggest that Ramadan fasting may be a suitable starting point as an intervention for the global obesity epidemic, including in many Muslim-majority countries. However, it would be important to further develop strategies to maintain any beneficial effects of the Ramadan period on obesity, after the fast is completed.

## Figures and Tables

**Figure 1 nutrients-11-00478-f001:**
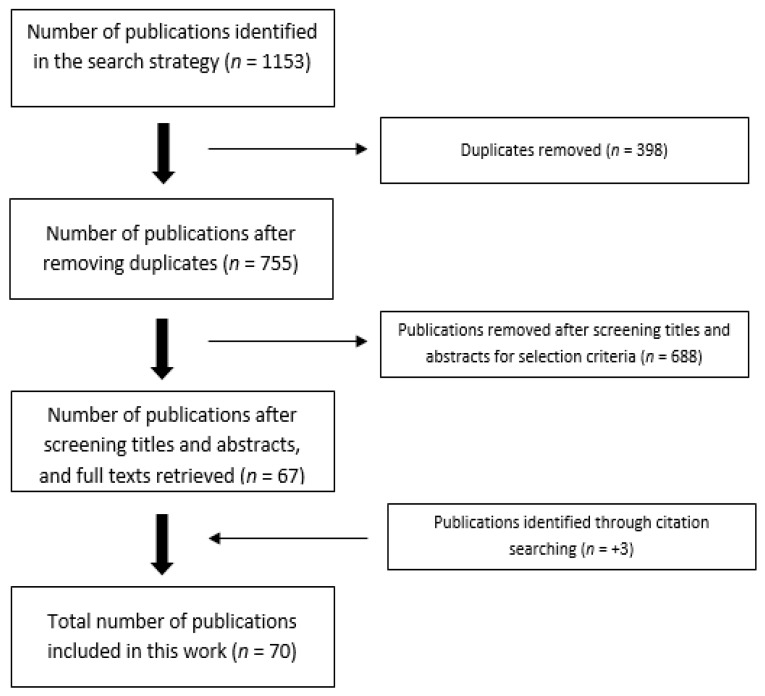
Flowchart for the selection of publications for systematic review and meta-analysis.

**Figure 2 nutrients-11-00478-f002:**
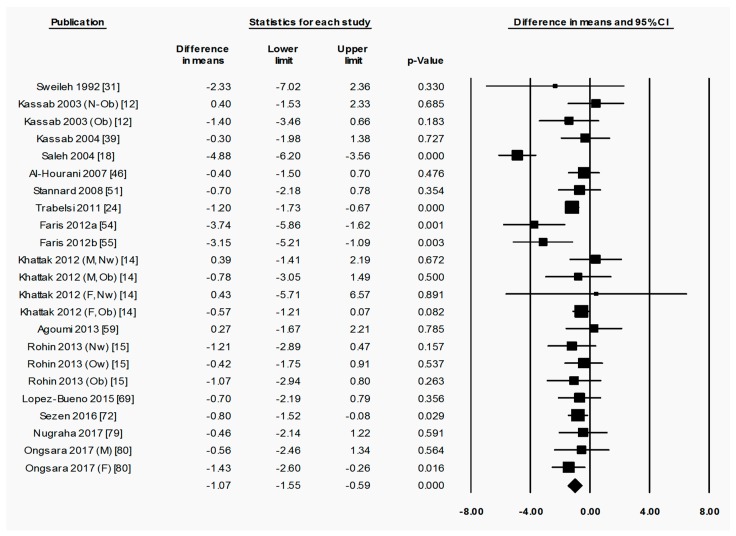
Fat percentage was significantly decreased by Ramadan fasting. Change in fat percentage (%) between pre-Ramadan (pre-R) and the end of Ramadan (post-R). Heterogeneity statistics: *T* = 0.829, *p* < 0.001, *I*^2^ = 61.5%. F = female; M = male; Nw = normal weight; Ow = overweight; Ob = obese; N-Ob = non-obese.

**Figure 3 nutrients-11-00478-f003:**
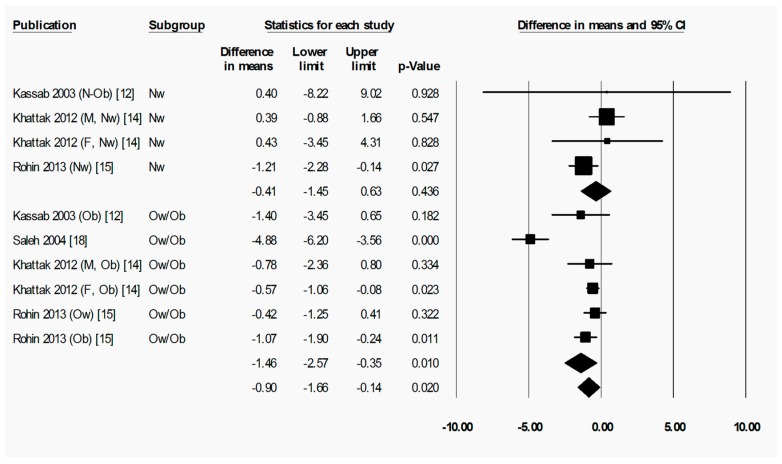
Fat percentage was significantly decreased by Ramadan fasting in the overweight/obese but not in the normal weight subgroup. Change in fat percentage (%) as subdivided by BMI category (normal weight and overweight/obese), between pre-Ramadan (pre-R) and the end of Ramadan (post-R). There was no significant difference between normal weight and overweight/obese subgroups (*p* = 0.178). Heterogeneity statistics: normal weight *T* = 0.508, *p* = 0.280, *I*^2^ = 21.7%; overweight/obese *T* = 1.536, *p* < 0.001, *I*^2^ = 86.8%. F = female; M = male; Nw = normal weight; Ow = overweight; Ob = obese; N-Ob = non-obese.

**Figure 4 nutrients-11-00478-f004:**
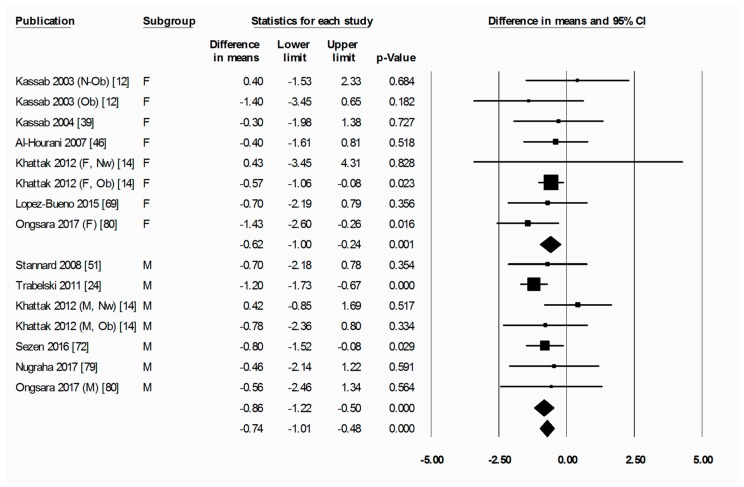
Fat percentage was significantly decreased by Ramadan fasting in both the female and the male subgroup. Change in fat percentage (%) as subdivided by sex (female and male), between pre-Ramadan (pre-R) and the end of Ramadan (post-R). There was no significant difference between female and male subgroups (*p* = 0.375). Heterogeneity statistics: female *T* = 0.000, *p* = 0.771, *I*^2^ = 0.0%; male *T* = 0.000, *p* = 0.435, *I*^2^ = 0.0%. F = female; M = male; Nw = normal weight; Ob = obese; N-Ob = non-obese.

**Figure 5 nutrients-11-00478-f005:**
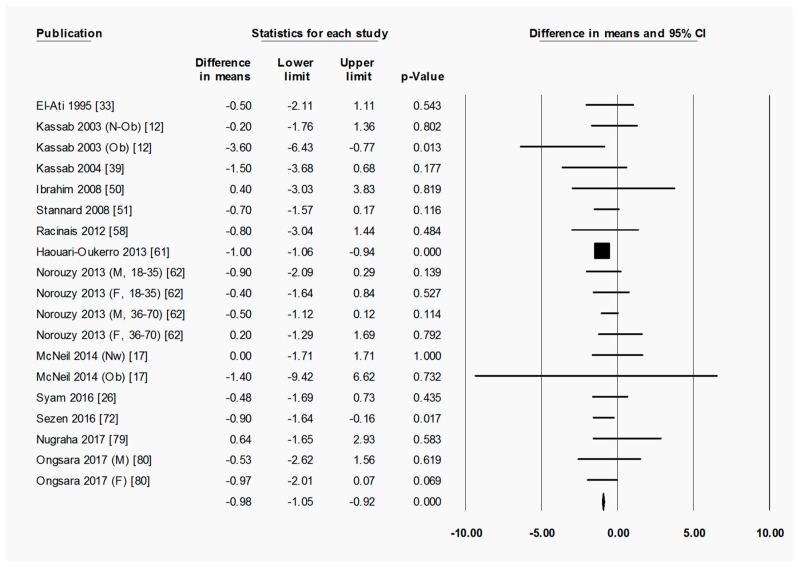
Absolute fat mass was significantly decreased by Ramadan fasting. Change in absolute fat mass (kg) between pre-Ramadan (pre-R) and the end of Ramadan (post-R). Heterogeneity statistics: *T* = 0.000, *p* = 0.600, *I*^2^ = 0.0%. F = female; M = male; Nw = normal weight; Ob = obese; N-Ob = non-obese; 18–35, 36–70 = age range.

**Figure 6 nutrients-11-00478-f006:**
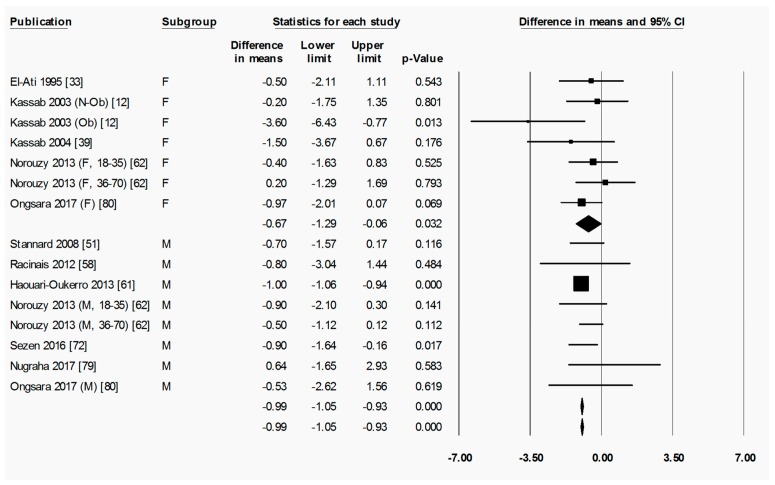
Absolute fat mass was significantly decreased by Ramadan fasting in both the female and the male subgroup. Change in absolute fat mass (kg) as subdivided by sex (female and male), between pre-Ramadan (pre-R) and the end of Ramadan (post-R). There was no significant difference between female and male subgroups (*p* = 0.314). Heterogeneity statistics: female *T* = 0.298, *p* = 0.333, *I*^2^ = 12.7%; male *T* = 0.000, *p* = 0.635, *I*^2^ = 0.0%. F = female; M = male; Ob = obese; N-Ob = non-obese; 18–35, 36–70 = age range.

**Figure 7 nutrients-11-00478-f007:**
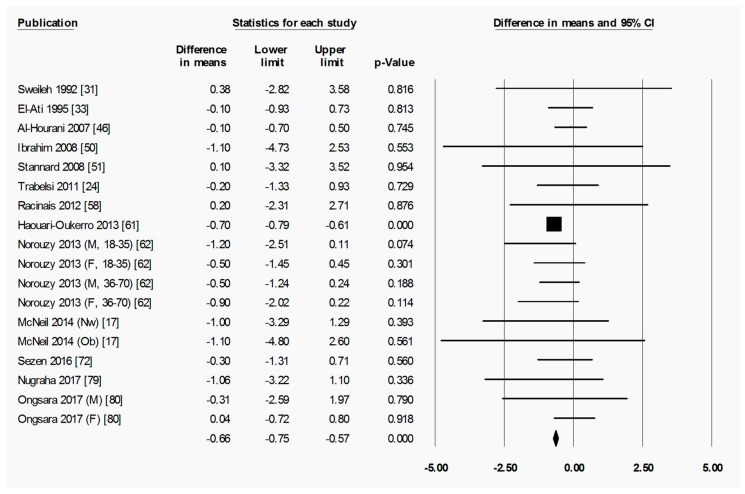
Fat-free mass was significantly decreased by Ramadan fasting. Change in fat-free mass (kg) between pre-Ramadan (pre-R) and the end of Ramadan (post-R). Heterogeneity statistics: *T* = 0.000, *p* = 0.753, *I*^2^ = 0.0%. F = female; M = male; Nw = normal weight; Ob = obese; 18–35, 36–70 = age range.

**Figure 8 nutrients-11-00478-f008:**
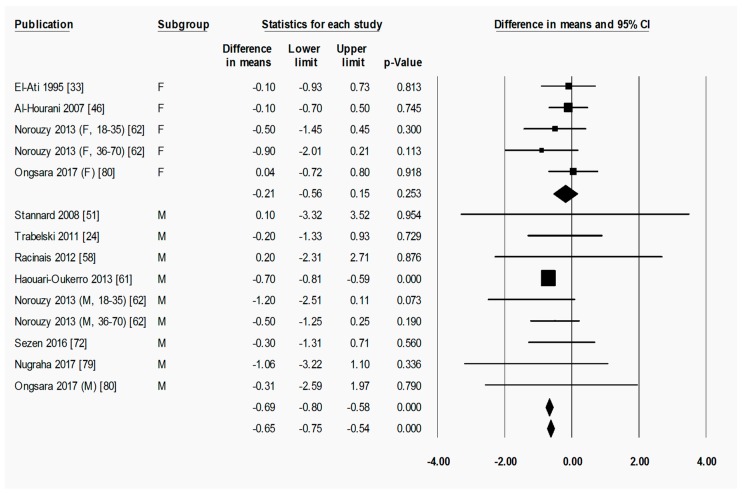
Fat-free mass was significantly decreased by Ramadan fasting in the male but not in the female subgroup. Change in fat-free mass (kg) as subdivided by sex (female and male), between pre-Ramadan (pre-R) and the end of Ramadan (post-R). There was no significant difference between female and male subgroups (*p* = 0.360). Heterogeneity statistics: female *T* = 0.360, *p* = 0.653, *I*^2^ = 0.0%; male *T* = 0.000, *p* = 0.948, *I*^2^ = 0.0%. F = female; M = male; 18–35, 36–70 = age range.

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
