# Peer review of "Effect of Ramadan Fasting on Weight and Body Composition in Healthy Non-Athlete Adults: A Systematic Review and Meta-Analysis"

_nutrients, 2019, doi:10.3390/nu11020478_

Round 1

Reviewer 1 Report

The effect of intermittent fasting on weight and  body composition is the subject of a growing body of scientific literature.

This is a well-written and thorough systematic review and meta-analysis examining the effect of Ramadan fasting on weight and body composition. The review follows the structure of the PRISMA guidelines. The findings are that Ramadan fasting is associated with significant decreases in weight, fat mass and fat-free mass. These effects were greater in subjects with higher baseline BMI. The effects appeared to be transient, with a return to previous values at 2-5 weeks post-Ramadan.

No significant issues were identified in review of this paper- although perhaps inclusion of the PRISMA checklist would be a useful addition. I think that this work contributes to the literature regarding intermittent fasting and the authors deserve acknowledgement of their efforts.

Author Response

Dr. Amy Xie

Editor

Nutrients

Saturday 16th February 2019

Re: Fernando et al. manuscript ID nutrients-433841

Dear Dr. Xie,

Thank you for your email dated 12th February 2019 providing us with the Reviewers’ comments on our manuscript nutrients-433841 entitled Effect of Ramadan fasting on weight and body composition in healthy non-athlete adults: a systematic review and meta-analysis. We are pleased that both Reviewers expressed interest in our study and had largely positively feedback.

Please find attached our revised manuscript in which we have addressed all of the comments offered by the two Reviewers as outlined in our point-by-point rebuttal below. The changes have been underlined in our revised manuscript and Supplementary materials.

We trust that the following satisfactorily addresses the Reviewers’ comments and that our revised manuscript is now suitable for publication in Nutrients.

Yours sincerely,

Professor Amanda Sainsbury-Salis

Detailed responses to the Reviewers’ comments

Reviewer #1

No significant issues were identified in review of this paper- although perhaps inclusion of the PRISMA checklist would be a useful addition. I think that this work contributes to the literature regarding intermittent fasting and the authors deserve acknowledgement of their efforts.

Our response: Thank you for the acknowledgement, we worked hard to optimise every step of this manuscript. We have now included the table for the PRISMA checklist in the Supplementary Materials. We have also mentioned this table towards the end of the Discussion of the main manuscript.

Reviewer #2

The formula for calculating the standard deviation of change should be highlighted from the text.

Our response: In the revised manuscript, there is now a dashed box around this formula, and a number to the left of it, as added by the Editorial Staff.

Unfortunately, I noticed in the references some discrepancies:

Please check citation #6, #27 (date should be added), #55, #68 and #75 - is this the same publication?

Our response: Thank you for the overall positive feedback. The mistakes with the references have now been corrected; we appreciate you noticing and pointing them out. As a result of the addition of the same reference twice under slightly different names, the overall results for weight and weight by location (as depicted by Figures S1 and S8, respectively, in the Supplementary materials) changed very slightly, with no effect on the conclusions. These have been described in the revised Manuscript within the Results section.

Reviewer 2 Report

The aim of the present systematic review and meta-analysis was to assess how Ramadan fasting affects weight and body composition in adults. Inclusion and exclusion criteria and the screening of the publications was described in detail in the materials and methods section of the manuscript.

The results part is very detailed, a sufficient number of graphs are shown.

The formula for calculating the standard deviation of change should be highlighted from the text.

In the final discussion section the relation to recent literature is discussed as well as strenghts and limitations are highlighted. 

Unfortunately, I noticed in the references some discrepancies:

Please check citation #6, #27 (date should be added), #55, #68 and #75 - is this the same publication?

Author Response

(The authors gave the same response as above.)
